# Statin Use in Relation to COVID-19 and Other Respiratory Infections: Muscle and Other Considerations

**DOI:** 10.3390/jcm12144659

**Published:** 2023-07-13

**Authors:** Beatrice A. Golomb, Jun Hee Han, Peter H. Langsjoen, Eero Dinkeloo, Alice E. Zemljic-Harpf

**Affiliations:** 1Department of Medicine, University of California, San Diego, La Jolla, CA 92093, USA; jhh002@health.ucsd.edu; 2Langjoen Cardiology Clinic, Tyler, TX 75701, USA; peterlangsjoen@cs.com; 3Navy and Marine Corps Public Health Center, Portsmouth, VA 23704, USA; edinkeloo@protonmail.com; 4Department of Anesthesiology, University of California, San Diego, La Jolla, CA 92093, USA; 5Veterans Affairs San Diego Healthcare System, San Diego, CA 92093, USA

**Keywords:** statin, COVID-19, muscle, rhabdomyolysis, long-COVID, post-COVID syndrome

## Abstract

Statins have been widely advocated for use in COVID-19 based on large favorable observational associations buttressed by theoretical expected benefits. However, past favorable associations of statins to pre-COVID-19 infection outcomes (also buttressed by theoretical benefits) were unsupported in meta-analysis of RCTs, RR = 1.00. Initial RCTs in COVID-19 appear to follow this trajectory. Healthy-user/tolerator effects and indication bias may explain these disparities. Moreover, cholesterol drops in proportion to infection severity, so less severely affected individuals may be selected for statin use, contributing to apparent favorable statin associations to outcomes. Cholesterol transports fat-soluble antioxidants and immune-protective vitamins. Statins impair mitochondrial function in those most reliant on coenzyme Q10 (a mevalonate pathway product also transported on cholesterol)—i.e., those with existing mitochondrial compromise, whom data suggest bear increased risks from both COVID-19 and from statins. Thus, statin risks of adverse outcomes are amplified in those patients at risk of poor COVID-19 outcomes—i.e., those in whom adjunctive statin therapy may most likely be given. High reported rates of rhabdomyolysis in hospitalized COVID-19 patients underscore the notion that statin-related risks as well as benefits must be considered. Advocacy for statins in COVID-19 should be suspended pending clear evidence of RCT benefits, with careful attention to risk modifiers.

## 1. Introduction

Since the COVID-19 pandemic broke in the U.S. in March 2020, numerous sources have directly advocated statin use for treatment of COVID-19, have provided evidence nominally supporting statin use for COVID-19, or have focused on observational associations of statin use with COVID-19 that might be construed as supporting statin use [1] (Appendix A). We do not reprise the arguments here, though we address the related observational studies. Points in favor of statins include their anti-inflammatory effects [2] and reduction in cardiovascular outcomes [3], outcomes that can arise in association with COVID-19 [4,5]. (Among additional hypothesized mechanisms of benefit include in silico evidence that some statins may affect the virus itself [5].) Widely disseminated position pieces have advocated while others have appeared to endorse off-label (high-dose) statins for COVID-19 infections by suggesting potential benefits of statins in this setting.

On the other side of the risk–benefit equation, we show evidence that adverse effects (AEs) of statins share mechanisms with AEs of COVID-19, and that many of these same adverse outcomes are present for both. Such AEs include, but are not limited to, muscle pain, muscle weakness, and fatigue. Rhabdomyolysis and autoimmune necrotizing myositis specifically have occurred with COVID-19 [6,7,8,9] as they can occur with statins [10,11,12,13,14,15,16]. Based on these and other considerations, the risk side of the risk–benefit ratio should also be considered seriously in deliberations about statin use. Of further note, key risk factors for adverse statin outcomes are also risk factors for adverse COVID-19 outcomes (e.g., hypertension, diabetes, obesity, and older age) [17], so those at high COVID-19 risk, in whom adjunctive therapy might be advised, are those in whom statins are most likely to produce serious problems.

Here, we review relevant evidence related to the use of statins for COVID-19 and post-COVID syndrome (long-COVID), addressing primarily those issues that have received little attention in the literature to date. We demonstrate that massive “healthy-user/healthy-tolerator” effects underlie favorable statin associations to infection. Although pre-COVID-19 observational studies showed apparent large favorable associations of statins to infection, meta-analysis of randomized controlled trials (RCTs) failed to support even a small causal effect. Observational associations for COVID-19 have included similar favorable associations, also these were not reproduced in randomized trial evidence. We address additional information related to effects of statins and functions of cholesterol that might inform deliberations about statin use in the setting of COVID-19, particularly in high-risk COVID-19 patients. Thus, we address clinical evidence related to statin use in COVID-19 and review underconsidered physiological and epidemiological factors that aid in evaluating and understanding this clinical evidence.

Key considerations extend to long-COVID. Symptoms of long-COVID prominently encompass symptoms that can arise with statins, including muscle weakness, fatigue, fatigue with exertion, and cognitive compromise [18,19,20,21,22,23].

## 2. Statins

Cholesterol-lowering medications, mostly statins, are used by approximately one in four persons over the age of 40 in the U.S. [24] (with widespread use elsewhere), adding high importance to understanding potential interactions between statin use and COVID-19 infection. Statins are pharmacological inhibitors of HMG-CoA reductase with regulatory effects on mevalonate synthesis. HMG-CoA reductase is ubiquitously expressed in mammalian cells and found in the membrane of the endoplasmic reticulum. Statins inhibit pathways for which mevalonate is a precursor (Figure 1), affecting not only synthesis of cholesterol and its downstream products, but multiple other pathways. Among these are the production of heme A and coenzyme Q10 (CoQ10, also known as ubiquinone), products of the inhibited mevalonate pathway that are required for mitochondrial electron transport which drives ATP synthesis. CoQ10 deficiencies (whether primary or secondary/acquired) may affect multiple organ systems (including heart and skeletal muscle, as well as other organs that can be affected by COVID-19—lungs, central and peripheral nervous systems, liver, and kidney) [25,26]. Inhibiting the mevalonate pathway, statins lead to dose-dependent reduction in CoQ10. CoQ10 is also an important endogenous antioxidant [27,28] that is transported in association with cholesterol [29]. Just as CoQ10 supplementation can effectively bypass mitochondrial “respiratory chain” defects [30], improving cell energy, so too can the lowering of CoQ10 with statins unmask (precipitate) mitochondrial compromise [31]—impairing cell energy and increasing free radical release [32,33]. These effects on mitochondrial energy production and impaired antioxidation contribute to statin AEs, including but not limited to risk of muscle symptoms, new-onset diabetes, and cognitive impairment [20,21,34,35].

The mevalonate pathway, for which statins inhibit the rate-limiting step, produces multiple compounds of relevance to statin effects as well as COVID-19 effects. Both coenzyme Q10 and heme A play critical roles in mitochondrial function. Mitochondrial function in turn moderates oxidative stress and resulting apoptosis, inflammation, and coagulation activation, blood flow/endothelial function, and autoimmunity. Testosterone is important for muscle strength and for COVID-19 cell entry. Other examples of these pathways also play roles in the muscle effects of statins, relevant in the COVID-19 setting. Statins have been reported to promote multiple pleiotropic mechanisms, including benefits to endothelial function, inflammation, coagulation activation, as well as antioxidation. However, antioxidation itself may be responsible for much or most of these benefits, since antioxidation supports endothelial function [36,37] and reduces apoptosis [38,39] that in turn promotes inflammation [40] and coagulation activation [40]. (In fact, inflammation and coagulation activation were strong predictors of severe COVID-19 outcomes [41,42], and this may, in part, arise from the fact that these reflect oxidatively mediated cell death.) A full discussion of pleiotropic effects of statins, the factors that underlie these, and the possible contribution of these to effects of statins on COVID-19 is outside the scope of this paper. The absence of randomized trial support for net statin benefit in treating patients with COVID-19 materially diminishes the imperative for detailed exposition of these pleiotropic mechanisms. Our focus is on key documented mechanisms with direct relevance for statins, for COVID-19, and for the interaction between them.

Statins are widely used to reduce low-density lipoprotein cholesterol (LDL) and to lower cardiovascular morbidity and mortality. Cardiovascular morbidity and cardiovascular mortality are reduced with statins in most statin trials, but effects on patient-centered (versus cause-specific) outcomes—those most relevant for preventive medications—show more mixed results, with a number of RCTs showing no trend to improvement in all-cause mortality or, where assessed, a proxy for all-cause serious morbidity (e.g., EXCEL, AFCAPS [43], ALLHAT-LLT [44], PROSPER [45]).

## 3. COVID-19 (SARS-CoV-2 Infection)

COVID-19 can be asymptomatic, can present with mild illness (e.g., with upper respiratory symptoms, fever, and fatigue), or can manifest severe disease encompassing refractory hypoxemia, pneumonia, and cytokine storm [46]. Severe disease can necessitate intensive care, can spur intubation (an issue depending on COVID-19 strain) and ventilator support, and can progress to death. Protean potential complications of COVID-19 can include (but are not limited to) cardiovascular [4], neurological [47], muscular [47], hepatic [48], and renal [48] compromise. 

Of note, clinical predictors of adverse COVID-19 outcomes largely encompass conditions that amplify (or reflect amplified) cell energy impairment: metabolic syndrome features (hypertension, obesity, diabetes), as well as older age [49]. Mitochondrial impairment rises exponentially with age [32] and, although this is underrecognized, metabolic syndrome features are markers of (and adaptations to) impaired cell energy [49]—often including mitochondrial impairment [49]. Other COVID-19 risk factors also reflect and/or contribute to impaired cell energy, including congestive heart failure (CHF) and chronic kidney disease [50] (in turn often tied to diabetes and/or hypertension). 

## 4. Statins and COVID-19 Outcomes: Observational Studies and Randomized Controlled Trials

Favorable associations of statins to respiratory and other infection outcomes before the COVID-19 pandemic, though of large magnitude, were subsequently shown to be chimerical, with no suggestion of favorable effect in meta-analysis of randomized trials [51,52].

Specifically, a pre-COVID-19 meta-analysis of a total of 20 observational studies states, “Meta-analysis for various infection-related outcomes revealed the following pooled odds ratios all in favor of statin use vs. non: 0.61 (95% confidence interval [CI], 0.48–0.73) for 30-day mortality (*n* = 7), 0.38 (95% CI, 0.13–0.64) for in-hospital mortality (*n* = 7), 0.63 (95% CI, 0.55–0.71) for pneumonia-related mortality (*n* = 7), 0.33 (95% CI, 0.09–0.75) for bacteremia-related mortality (*n* = 4), 0.40 (95% CI, 0.23–0.57) for sepsis-related mortality (*n* = 4), and 0.50 (95% CI, 0.18–0.83) for mixed infection-related mortality (*n* = 4)” [51]. These pooled estimates encompass individual studies that report still more favorable associations, which we note in order to point out, in juxtaposition with the RCT meta-analysis to follow, that even apparently massive favorable associations have no necessary implications for causal effect occurrence.

However, a subsequent meta-analysis of RCTs examining the actual causal relation of statin use to such outcomes found no relationship. A meta-analysis showed no effect of statins on the risk of infections or on infection-related deaths: relative risk (95% confidence interval, or CI), 1.00 (0.96–1.05) and 0.97 (0.83–1.13), respectively [52]. Anti-inflammatory benefits of statins did not translate to clinical benefits to infection, when the best quality evidence was considered. The disparity between massive favorable observational relationships (associations) and no relationships (effects) whatsoever on RCTs comports with the often-seen, well-documented large “healthy-user” and “healthy-tolerator” effects for statins (and other preventive medicines) spanning many outcomes [53]. Indeed, the chasm between observational and randomized findings is evident in that observational associations showed apparent “reductions” (favorable associations) in some infection-related outcomes exceeding 60% upon pooled analysis [53]. There may also be a role for “indication bias”, in which statin use serves as a proxy for higher cholesterol (prior to or during COVID-19 infection—see below) [53].

Similar associations are expected for statin use with COVID-19 outcomes, and indeed analogous favorable associations have now been reported [54,55,56]. (See Table 1 for a table summarizing the following results.) In one analysis, use of statins prior to admission was tied to reduced risk of severe COVID-19: adjusted odds ratio (OR) (95% CI), 0.29 (0.11–0.71), *p* < 0.01 [54]. In another analysis, statin use was associated with favorable outcomes, in a proportional hazards model adjusted for baseline differences, considering statins as a time-varying exposure: adjusted hazard ratio (95% CI), 0.63 (0.48–0.84, *p* = 0.001). In a mixed-effects Cox model that did not consider statins as a time-varying exposure, statin use was also associated with a decreased incidence of death, OR, 0.58 (0.43–0.80, *p* = 0.001) [55]. Sequential meta-analyses reported the following (none of which de facto included randomized trials), with the third meta-analysis being a follow-up to the first, incorporating subsequently published studies [57,58,59]: total included number of trials in the three were *n* = 24, *n* = 22, and *n* = 47, respectively. Each of the studies provided data on in-hospital COVID-19 mortality both for pre-hospital statin use and in-hospital statin use. For pre-hospital statin use, the mortality ORs/relative risks (95% CIs) were: 0.77 (0.60–0.98), 0.69 (0.56–0.84), and 1.06 (0.82–1.37) based on *n* = 18, *n* = 14, *n* = 29 studies, respectively. For in-hospital statin use, the respective values for the three meta-analyses were: 0.40 (0.22–0.73), 0.57 (0.54–0.60), and 0.54 (0.50–0.58) based on *n* = 3, *n* = 11, and *n* = 7 included studies, respectively. A further “meta-analysis” published as a letter to the editor (that included two of our four peer-reviewed citations, plus two non-peer-reviewed findings), showed a pooled hazard ratio for “fatal or severe” COVID-19 outcomes: 0.70 (0.53–0.94) [60]. (Subsequent observational studies of statin use in hospitalized COVID-19 patients show analogous favorable associations to severe and fatal COVID-19 outcomes [61,62].) However, in-hospital statin use is expected to be a proxy for higher in-hospital LDL cholesterol, which is expected to be a proxy for better expected outcomes (irrespective of statin use) in two different ways. First, higher pre-infection LDL is correlated with higher in-hospital LDL, other events being equal; as discussed elsewhere, higher pre-hospital cholesterol is tied to lower risk of severe infection outcomes. Second, since acute infection lowers cholesterol, with cholesterol reduction correlated with infection severity [63], higher in-hospital cholesterol (and corresponding comfort with administration of statins) is associated with less severe COVID-19 infection and expectation of lower COVID-19 mortality. Thus, in-hospital statin use—through its presumed relation to higher in-hospital LDL—serves as a proxy for lower prior risk of, and lower current presence of, severe infection. Although the authors do not mention this, a large favorable association of in-hospital statin use to lower mortality has no necessary implications for whether the causal effect of in-hospital statin use on mortality in hospitalized COVID-19 patients is favorable, adverse, or neutral.

Not all observational studies, even those of in-hospital statin use, report favorable associations. An Italian study including 3988 COVID-19 patients showed no independent statin association to mortality upon multivariable analysis: hazard ratio (95% CI), 0.98 (0.81–1.20) [64].

A sizable study of close to 20,000 patients with COVID-19 found that statin use was not associated with improved outcomes (encompassing outcomes like death and ICU stay, according to what was assessed in the encompassed study in a meta-analysis) in unadjusted analysis: OR (95% CI), 1.02 (0.69–1.50, *p* = 0.94). However, statin use was associated with markedly reduced risk of adverse outcomes with adjusted analysis: OR (95% CI), 0.51 (0.41–0.63, *p* < 0.0005) [65]. Other observational studies variably report beneficial associations to symptoms (but not serious outcomes) [66] and beneficial associations to severe COVID-19 and faster time to recovery [54]. Other studies showed no significant association or a significantly adverse association to severe COVID-19 and/or mortality. A meta-analysis of nine studies showed no association between statin use and severe outcomes (OR (95% CI), 1.64 (0.51–5.23, *p* = 0.41)) or mortality (OR, 0.78 (0.50–1.21, *p* = 0.26)) [67]. A Johns Hopkins study showed no association between statin use and mortality in hospitalized patients (relative risk (95% CI), 1.00 (0.99–1.01, *p* = 0.928)) and an association to increased severe COVID-19 (relative risk, 1.18 (1.11–1.27, *p* < 0.001)) [68]. Finally, in a study of COVID-19 mortality in patients with type 2 diabetes with an average age of 70.9, statin use was associated with significantly increased mortality at 7 days (OR (95% CI), 1.74 (1.13–2.65)) and at 28 days (OR, 1.46 (1.08–1.95)) [69].

**Table 1 jcm-12-04659-t001:** Observational studies and meta-analyses of statin use in the COVID-19 setting.

First Author	N	Population	% on Statins	Outcomes	Measure	Results	95% CI	*p*
Daniels et al. (2020) [54]	170	Adult patients hospitalized with COVID-19, at UC San Diego Health	27%, on admission	Reduced risk of severe COVID-19	OR	0.29	0.11–0.71	*p* < 0.01
				Faster time to recovery among those without severe disease	HR	2.69	1.36–5.33	*p* < 0.01
Zhang et al. (2020) [55]	13,981	Adult hospitalized patients with COVID-19, in Hubei Province, China	8.7%	Cox time-varying model: All-cause mortality	HR	0.63	0.48–0.84	*p* = 0.001
				Mixed-effects Cox model: All-cause mortality		0.58	0.43–0.80	*p* = 0.001
Grasselli et al. (2020) [64]	3988	Adult hospitalized patients with COVID-19, in Italy	12%	Multivariable Cox proportional hazards regression analysis: mortality	HR	0.98	0.81–1.20	*p* = 0.87
Ayeh et al. (2021) [68]	4447	Adult hospitalized patients with COVID-19, at John Hopkins Medical Institutions	13.4%	Mortality	RR	1.00	0.99–1.01	*p* = 0.928
				Severe Infection		1.18	1.11–1.27	*p* < 0.001
Cariou et al. (2021) [69]	2449	Adult hospitalized patients with COVID-19 and Type 2 Diabetes, from CORONADO study	49%	7-day mortality	OR	1.74	1.13–2.65	N/A
				28-day mortality		1.46	1.08–1.95	
Israel et al. (2020) [56]	6530	Adult hospitalized patients with COVID-19, in Israel	5.0%	Hospitalization	OR	0.673	0.596–0.758	*p* < 0.001
Meta-analyses								
Vahedian-Azimi et al. (2021) [57]	32,715 (24 studies total)	Varies by study	Varies by study	Pre-hospital use of statins: mortality (*n* = 18 studies)	OR	0.77	0.60–0.98	N/A
				In-hospital use of statins: mortality (*n* = 3 studies)		0.40	0.22–0.73	
Wu et al. (2021) [58]	(22 studies total)	Varies by study	Varies by study	Pre-hospital use of statins: mortality	RR	0.69	0.56–0.84	*p* < 0.001
				In-hospital use of statins: mortality		0.57	0.54–0.60	*p* < 0.001
Vahedian-Azimi et al. (2021)—Follow-up [59]	3,238,508 (47 studies total)	Varies by study	Varies by study	Pre-hospital use of statins: mortality (*n* = 29 studies)	OR	1.06	0.82–1.37	*p* = 0.670
				In-hospital use of statins: mortality (*n* = 7 studies)		0.54	0.50–0.58	*p* < 0.001
Kow et al. (2020) [60]	Four studies	Varies by study	Varies by study	“Fatal or severe” COVID-19 outcomes	HR	0.70	0.53–0.94	N/A
Pal et al. (2021) [65]	19,988 (14 studies total)	Varies by study	Varies by study	Unadjusted analysis: statin use not associated with improved clinical outcomes	OR	1.02	0.69–1.50	*p* = 0.94
				Adjusted analysis: statin found to reduce risk of adverse outcomes		0.51	0.41–-0.63	*p* < 0.0005
Hariyanto et al. (2020) [67]	3449 (Nine studies total)	Varies by study	Varies by study	Severe outcomes	OR	1.64	0.51–5.23	*p* = 0.41
				Mortality		0.78	0.50–1.21	*p* = 0.26

OR = odds ratio; HR = hazard ratio; RR = relative risk/risk ratio.

Another study examined drugs associated with reduced COVID-19 severity. Associations for rosuvastatin with reduced hospitalization were consistent with aforementioned healthy-user/tolerator effects: OR (95% CI), 0.673 (0.596–0.758, *p* < 0.001) for COVID-19 hospitalization [56]. Of interest was the far larger favorable association of ubiquinone (coenzyme Q10), which is not expected to have similar healthy-tolerator contributions: OR (95% CI), 0.185 (0.058–0.458). The OR for hospitalization of <0.19 is noteworthy since statins lead to dose-dependent reductions in coenzyme Q10, which is also a product of the mevalonate pathway and is transported with cholesterol [35]. 

On a relevant note, given the occurrence of acute respiratory distress syndrome (ARDS) in COVID-19 patients, it is of interest that a retrospective analysis of an RCT including 683 subjects with infection-related acute respiratory distress syndrome (ARDS) found that rosuvastatin administration was associated with increased interleukin-18 plasma levels, which were tied to higher mortality [70]. Consistent with this, in mouse models, statin exposure also increased inflammasome activation, interleukin-18 expression, and acute lung injury [71]. 

Four randomized controlled trials of statin use in the COVID-19 setting were identified [72,73,74,75]. (See Table 2 for a table summarizing the following results.) A search of PubMed was conducted on 18 April 2022 through EndNote using the following search terms as title words: “statin or rosuvastatin or atorvastatin or fluvastatin or simvastatin or lovastatin or pravastatin” and “COVID or SARS-CoV-2” and “randomized”. An expanded search was attempted, allowing “randomized” to be an abstract term, but no additional randomized trials were identified with this approach. A subsequent search (October 2022) identified one additional RCT [75]. These findings are at best mixed, with one apparently unequivocally adverse result, two neutral results, and one result favorable for a single outcome out of a suite of assessed outcomes. Given these findings, existing RCT data do not provide support for statin use in the acute COVID-19 setting. A fifth randomized trial included statin together with colchicine but did not separate the effects of these two agents and so can provide no information on the independent effects of statins [76]. The four identified RCTs have findings as follows: The first RCT, with a sample size of *n* = 156, randomized hospitalized COVID-19 patients to “standard therapy” (hydroxychloroquine + Kaletra^®^) or 20 mg of atorvastatin + standard therapy. Statin vs. placebo led to the following reported effects: mean hospitalization days (7.72 days vs. 5.06 days, *p* = 0.001) and frequency of hospitalization in the ICU (18.4% vs. 1.3%, *p* = 0.001) were longer and greater in those randomized to atorvastatin compared to the comparison group [72]. The second (*n* = 587) randomized patients to 20 mg of atorvastatin or placebo and reported that atorvastatin was not associated with a significant reduction in the primary outcome, which was a composite of venous or arterial thrombosis, treatment with extracorporeal membrane oxygenation, or all-cause mortality—OR (95% CI), 0.84 (0.58–1.21) [73]. The third (*n* = 40) randomized patients to 40 mg of atorvastatin + 400/100 mg of lopinavir/ritonavir or 400/100 mg of lopinavir/ritonavir alone, with outcomes as follows: duration of hospitalization (in days) was longer in the lopinavir/ritonavir group compared to the lopinavir/ritonavir + atorvastatin group (9.75 ± 2.29 vs. 7.95 ± 2.04, *p* = 0.012); however, there was no significant difference in the secondary outcomes (need for interferon or immunoglobulin and receipt of invasive mechanical ventilation) between the randomized groups [74]. The corresponding centers make use of other COVID-19 treatments and involve different patient populations, so that different interaction effects or sources of effect modification are not excluded. The fourth was a randomized, factorial design, open-label study of hospitalized patients with mild-to-moderate COVID-19 in which participants received atorvastatin 40 mg, aspirin 75 mg, both, or neither (*n* = 900, with *n* = 225 in the atorvastatin and both groups, *n* = 226 in the neither group). The primary outcome was “clinical deterioration to WHO Ordinal Scale for Clinical Improvement ≥ 6”. The rate of this outcome in the standard care group was 3.2%. In the statin group, it was also 3.2%: HR (95% CI), 1.0 (0.41–2.46, *p* = 0.99) [75]. (In the aspirin group, the rate was 1.4%; in the atorvastatin + aspirin group, it was 3.6%—neither differed significantly from the standard care group.) A 2023 meta-analysis of statin use in adult hospitalized COVID-19 patients identified the same four trials we previously identified and affirmed our conclusion: “There was no significant difference in all-cause mortality (odds ratio [OR] 0.96; 95% confidence interval [95% CI]: 0.61–1.51; *p* = 0.86…), duration of hospitalization (mean difference [MD] 0.21; 95% CI: −1.74–2.16; *p* = 0.83…), intensive care unit admission (OR = 3.31; 95% CI: 0.13–87.1; *p* = 0.47…), need for mechanical ventilation (OR = 1.03; 95% CI: 0.36–2.94; *p* = 0.95…)…between patients treated with or without statin therapy [77]”. Clearly, available randomized trial data do not provide meaningful support for statin administration for COVID-19 in patients. We underscore again the disparity between the large observational associations of in-hospital statin use to lower COVID-19 mortality upon meta-analysis and the absence of evidence for mortality improvement with in-hospital COVID-19 for patients in RCTs. These findings suggest that favorable observational associations of statins to COVID-19 outcomes (however large)—just as was seen above for favorable statin associations to pre-COVID-19 infection outcomes—have no implications for causal effects.

Incidentally, as treatments for COVID-19 emerge, risk–benefit considerations for statins may further alter, in particular, due to potential drug interactions. For instance, nirmatrelvir/ritonavir (“Paxlovid”), by inhibiting the CYP3A4 metabolizing pathway, may heighten risks associated with statins in proportion to the degree to which they are metabolized through that pathway. This especially affects simvastatin and lovastatin, which should not be co-administered, with potential interactions for atorvastatin and rosuvastatin for which close monitoring and potential dose adjustment are advised [78].

## 5. Cholesterol and Infection

Higher cholesterol has been associated with lower risk of infection and respiratory disease outcomes, including pneumonia/influenza hospitalizations [79], respiratory disease deaths [80], and even HIV [81]. These associations examine long-term risk and are not driven by existing disease. Indeed, higher cholesterol measured even years before the HIV epidemic “broke” were tied to lower risk of death from AIDS later [81]. Similarly, higher cholesterol was tied to lower risk of death from respiratory diseases, even after omitting the first five years of deaths [80]. (That said, serious infection can also lower cholesterol, which may contribute to the lower cholesterol reported in COVID-19 patients in China [82].)

Functions of cholesterol associated with infection protection are beyond the scope of this review but, briefly, include transport of nutrients that are critical to mucosal immunity, to innate and adaptive immunity, to quelling of inflammation in the setting of infection, to support for cell energy (important in COVID-19, where there is low energy supply, e.g., due to hypoxemia and high energy demand), and confer critical protection against oxidative stress (including from reoxygenation injury arising from COVID-19 hypoxemia) which in turn can trigger apoptosis [83] and result in inflammation and coagulation activation [40] (see VI). Nutrients transported by cholesterol—and specifically by LDL—as well as products of these, include vitamins that have been shown in RCTs to protect against viral infection occurrence and mortality. Oxidative stress, against which these nutrients defend, also depresses natural killer cell number and function, amplifying viral infection risk. Cholesterol itself also plays a pivotal role in cell barrier function and in cell energy defense, the latter in part by reducing heavy energy demands of the transmembrane sodium/potassium gradient.

Cholesterol declines more strongly with greater COVID-19 severity, as shown in multiple studies and meta-analyses, including for total cholesterol, LDL, and HDL cholesterol [84,85,86,87,88]. These drops in lipids not only reflect severity of disease, but because both LDL and HDL are involved in transporting critical fat-soluble antioxidants [89,90,91,92,93], and HDL also transports substances like paraoxonase with antioxidant (and anti-SARS-CoV-2 properties [94]) properties [95,96,97], these reductions might not just reflect but may conceivably contribute to worse COVID-19-associated outcomes. Statin use, clinically, is typically tied to/driven by levels of LDL cholesterol, which reproducibly respond to statin therapy [98]. HDL cholesterol is often, on average, neutrally influenced by statin therapy [99]. For these reasons, although HDL is of vital importance as a reflector of and contributor to oxidative stress defense potential, it is not the recipient of equal attention in the discussion here.

What about for COVID-19? An analysis of 5279 COVID-19 patients in New York City found that, in multivariable regression examining risk factors for hospitalization, diagnosed hyperlipidemia appeared protective: OR (95% CI), 0.62 (0.52–0.74, *p* < 0.001) [50]. (This does not imply that the lipids measured would remain elevated throughout the COVID-19 course, because of the large impact that severe infection can have in depressing cholesterol values [88].)

Long-term rates of community-acquired sepsis are greater specifically with lower LDL [100], suggesting that associations of higher cholesterol to lower infectious complications extend to bacterial conditions, which can complicate COVID-19. Statins can also have antioxidant effects; however, they are reproducibly prooxidant in some, and it is precisely the groups at high risk for COVID-19 outcomes (as below) in which this is expected.

## 6. Statins in COVID-19 High-Risk Groups

Potentially offsetting the reduction in cholesterol-associated transport of nutrients critical to viral defense, statins are often antioxidant. However, statins are sometimes reproducibly prooxidant [101]. This prooxidant predominance is tied to statin adverse effects [102], tied in turn to older age and metabolic syndrome factors [31]. It is in groups with worse mitochondrial status—e.g., those with older age or metabolic syndrome features—that CoQ10 withdrawal due to statins unmasks mitochondrial impairment [30] and increases free radical release [32,33]—offering prospects for statins’ prooxidant properties to override statins’ antioxidant mechanisms. 

Thus, in obese, diabetic, hypertensive, and/or older patients—patients at risk for serious COVID-19 outcomes [17] that might be deemed to most merit adjunctive off-label statin therapy—statins may be adverse not just to oxidative stress but, in consequence, to downstream effects of oxidative stress that may worsen COVID-19 sequelae as well. One relevant effect (and cause) of oxidative stress is mitochondrial dysfunction [33], for which adverse portent is magnified in patients with hypoxemia. Another, as above, is apoptosis [83], with resultant triggering of inflammation [40] and coagulation activation [40], two serious adverse prognostic indicators in COVID-19 [50]. (Though statins are often anti-inflammatory, effects can be bidirectional as with oxidative stress. As citations here show, statins are sometimes pro-inflammatory [103,104].)

The statin AEs to which these COVID-19 high-vulnerability groups are at heightened risk include cardiac, neurological, muscular, hepatic, and renal AEs [31]—shared domains with COVID-19 AEs, affording further reason for caution with statin administration in the setting of COVID-19. However, the best known statin AEs affect muscle [21,31,105]. 

## 7. Statins and Muscle: Airways and Heart

Skeletal muscle symptoms are the most commonly reported statin adverse effects [31], and COVID-19 (as well as long-COVID) is also tied to muscle complications [22,106,107,108,109,110]. As above, statin-associated reductions in CoQ10, unmasking mitochondrial compromise, can impair cell energy and increase free radical release [32,33]. A common consequence of impaired cell energy is impaired muscle strength [111,112,113]. COVID-19 also impairs cell energy via sometimes-profound hypoxemia [50], and low CoQ10 is also seen following COVID-19 [114]. Skeletal muscle symptoms, with CK elevation, are reported as a complication of COVID-19 [47], as they are of statins (see later discussion of rhabdomyolysis) [21,31,111]. Cardiac muscle complications, in the form of CHF, have been reported in association with statins [115,116] and are also reported with COVID-19 [117,118].

### 7.1. Skeletal Muscle (Airway Dilator Muscles and Muscles of Respiration)

Statins can increase muscle-related symptoms [21,31,105] including exertional fatigue [18] and muscle weakness [19,21] (with implications also for upper airway dilator muscles and muscles of respiration, particularly important in severe COVID-19 infection). Statins are particularly apt to produce such muscle compromise in older age [19,31] and presence of metabolic syndrome features [31]. Muscle weakness, from either mitochondrial impairment or hypoxemia, can compromise upper airway dilator muscles and promote airway narrowing especially in the supine position, promoting sleep apnea. For this reason, sleep apnea is common in mitochondrial disorders [119] and in other conditions of muscle weakness [119]. Presumably, it is via these means that statins have been reported to promote sleep apnea in some [120]. Impaired cell energy can compromise airway caliber and worsen hypoxemia, magnifying morbidity and mortality in settings of severe respiratory compromise—just as improved cell energy can contribute to the widening of airways (greater airway dilation), extending to daytime and non-supine positioning [121]. Airway dilator collapse is most prominent in the supine position. Apropos of this are representations of benefit to COVID-19 patients with “proning” and nasal continuous positive airway pressure (CPAP) therapy [122,123]. (Potentially underscoring the importance of airway integrity in COVID-19 outcomes, COVID-19 risk is particularly great in obese patients [124], in whom sleep apnea is singularly common.) Of note, sleep apnea is attended causally by oxidative stress [125,126,127,128] (related to hypoxemia and reoxygenation [126]), with oxidative stress being a potentially serious mechanism in COVID-19 [129,130,131,132,133], contributing to other factors that adversely affect cell energy such as endothelial function compromise [125,129,134]. (Like other oxidative stress-promoting exposures, bidirectional effects of hypoxemia on outcomes may arise in some settings [135], likely via the impact of oxidative preconditioning whereby modest exposure to an oxidative stressor may lead to upregulation of antioxidant defenses [136]—where these are not already overwhelmed—leading to favorable clinical effects. Oxidative preconditioning has been proposed as a potential therapeutic approach for COVID-19 [137].)

An additional consideration is the potential for both statins and COVID-19 to promote autoimmunity. Autoimmune necrotizing myositis [10,11,12,138] is a feared complication of statin use. (Other autoimmune complications of statins are also reported [139,140,141,142,143]—which can affect muscle [144,145,146,147]—and in some cases, these accompany autoimmune myositis [13,14,15,16,148].) Autoimmunity is also a complication of COVID-19 [149,150,151,152] (encompassing, specifically, autoimmune conditions reported with statins [6,7,153,154,155] including necrotizing myositis [8,9]) and is a factor in long-COVID [156,157,158,159]: in one “multi-omics” study of long-COVID, an autoimmune profile represented one of the four key patterns observed [160]. (See Section 11 for further discussion of long-COVID.)

### 7.2. Cardiac Muscle

Mitochondria comprise about 40% of the volume of cardiac muscle cells [161]. The heart consumes 20 times its weight in ATP per day [162]. Diastolic function, even more than systolic function, is strongly ATP-dependent [115]. Consistent with statin-induced CoQ10 depression [163], clinical findings suggest that statins may worsen diastolic function [115] and may induce statin-associated cardiomyopathy in vulnerable patients [116]. Conversely, in a double-blind RCT, CoQ10 supplementation cut all-cause mortality by 42% (*p* = 0.036) in heart failure patients [28]—underscoring the importance of CoQ10 alteration to heart failure outcomes. CoQ10 transport is among relevant functions of cholesterol-carrying lipoproteins [91] (in addition to CoQ10 being a product of the mevalonate pathway inhibited by statins). Therefore, risking worsening cardiac function with statin-induced mevalonate inhibition and CoQ10 depletion may be imprudent in those with factors prognostic for adverse COVID-19 outcomes—which as above, double as risk factors for adverse statin outcomes. (Concordant with these reports, atorvastatin exposure impaired the structure and function of cardiac mitochondria, thereby worsening cardiac outcomes in a mouse heart failure model [164].) CHF is an independent risk factor for adverse COVID-19 outcomes [4,50]—as well as a complication of COVID-19 [118]. Hypoxemia from COVID-19—like mitochondrial impairment from statins—can worsen CHF outcomes. 

Of note, low levels of serum total cholesterol and LDL are associated with reduced survival in heart failure patients [165]. This so-called “reverse epidemiology”, which occurs in a range of conditions tied to impaired cell energy [49], is evidently applicable irrespective of statin treatment [166]. 

## 8. Statins and COVID-19: Effects on Liver

Abundant evidence supports occasional hepatotoxicity with statin use. Although effects are commonly mild, more serious hepatotoxicity can occur [140,142,167,168]. Statin-mediated liver toxicity mechanisms likely include mitochondrial impairment [169] and extend to autoimmune hepatitis [139,141].

Similarly, COVID-19 confers potential for hepatotoxicity, mild or severe [170,171,172]. As with statins, mechanisms of liver toxicity likely include mitochondrial impairment [171] and also extend to autoimmune hepatitis [153,154,173,174].

## 9. Statin Effects on Testosterone: Implications for Muscle, as Well as Viral Entry to Cells, in Acute COVID-19

Statins lead to significant but highly variable depressions in testosterone (a biochemical product of cholesterol that, like all steroid hormones, is made in the mitochondria [175]). These effects, while significant on average in sufficiently sized samples or meta-analyses [176,177] and/or studies with higher potency statins [178], are highly variable. Since COVID-19 entry into cells is androgen-dependent (believed to contribute to the male preponderance of severe COVID-19) [179,180], this mechanism might confer benefit in relation to acute COVID-19. At the same time, our preliminary data show that the degree of the drop in testosterone in statin users is related to the degree of increase in muscle pain and muscle weakness (in data from a double-blind randomized controlled trial [181]), which could have implications for caliber/patency of the upper airway (through effects on upper airway dilator muscles [121]) and strength of respiratory musculature (and potentially cardiac muscle action).

## 10. Rhabdomyolysis

Rhabdomyolysis is promoted in settings and with exposures and conditions tied to impaired energy supply and/or increased energy demand—particularly, these risk factors in combination [31,182]. Statin-induced rhabdomyolysis (or, more generally, myopathy) is more common with greater statin potency [31] (which is more likely to impair cell energy) [31,163]. Risk is also greater in older age and in those with conditions like hypertension, obesity, and diabetes [31]. These are settings tied to impaired energy supply [49] and are also risk factors for worse outcomes (e.g., hospitalization, death) with COVID-19 [50,183]—settings where off-label adjunctive therapy might be tried.

Additional risk of rhabdomyolysis from statins in COVID-19 patients may emanate from drug interactions [31,184]. Drug interactions may arise with agents used for comorbidities and hospital-related prophylaxis. These may include recognized or yet-to-be-recognized statin interactions with agents used to address COVID-19 or its complications and interactions with antibiotics that may be given for known or suspected coinfections [185].

Moreover, infection is itself a reported risk factor for rhabdomyolysis [186,187]. The associated increase in energy demand [49,188] adds to risk of rhabdomyolysis with statin use [31] in high-risk COVID-19 patients. COVID-19, specifically, has characteristics that further magnify risk. Hypoxemia is common. Moreover, underlying mechanisms encompass oxidative stress (potentiating endothelial/blood flow impairment and mitochondrial impairment) and mitochondrial dysfunction [189] itself. That is, mechanisms implicated in COVID-19 severity parallel those of statin toxicity, explaining why risk factors are so tightly shared and underscoring the potential for statin use to compound risk of rhabdomyolysis in association with COVID-19.

What is the magnitude of threat of rhabdomyolysis in association with COVID-19? Retrospective cohort studies suggest it is unexpectedly high. Reported fractions of hospitalized COVID-19 patients who met criteria for rhabdomyolysis are as high as 16% and 20% [190,191]. (Rhabdomyolysis commonly involves multiple predisposing factors, so other risk factors besides COVID-19 were evident in a number of such cases.) Moreover, mortality of hospitalized COVID-19 patients with rhabdomyolysis may be extreme, with approximately half (47.1%) of such patients dying, in one series [191].

Beyond the hundreds of cases of COVID-19-associated rhabdomyolysis reflected in published retrospective reports of hospitalized COVID-19 patients, more than 100 individually reported cases of COVID-19-associated rhabdomyolysis have been published as case reports (to date). A good number involve concurrent statin use, consistent with ~30% of hospitalized rhabdomyolysis patients in the above retrospective study [191] taking statins prior to admission. Whether adjunctive use of statins for COVID-19 patients once hospitalized was a factor in any is not stated. Given the striking reported risk of rhabdomyolysis in hospitalized COVID-19 patients and the dire mortality prospects in such patients, the injudiciousness of introducing a further rhabdomyolysis risk factor (statins or other) imposes unwarranted risk.

## 11. Statin Risk–Benefit in Groups at High Risk for Adverse COVID-19 Outcomes

Above, we examined implications of statins related to COVID-19, for high-risk COVID-19 populations. We then focused on muscle implications. We now expand out, for a broader view of statin risk–benefit in patients at risk for adverse COVID-19 outcomes, addressing risk–benefit implications independent of COVID-19. Statins are less favorable—in terms of risk–benefit to patient-centered outcomes that balance serious risk and benefit, like all-cause mortality—in high-risk COVID-19 groups like the elderly (PROSPER) [45], those with hypertension (ALLHAT-LLT) [44], or other metabolic syndrome factors such as low high-density lipoprotein cholesterol (AFCAPS) [43]. The statin trials mentioned targeting these groups showed no trend to benefit to all-cause mortality or, where assessed, to a proxy for all-cause serious morbidity. This failure of statins to improve patient-centered outcomes in these trials of the elderly and of patients with these metabolic syndrome features, despite their elevated coronary disease risk, merits serious consideration in deliberations regarding use of statins in these groups—even in those without COVID-19 infection and, a fortiori, in those with COVID-19. The importance of this is underscored by the fact that, as above, such patients—precisely because of their high COVID-19 risk—may be prioritized for adjunctive statin therapy.

## 12. Statins and Long-COVID, Also Known as Post-COVID Syndrome

Statin use has been advocated by physicians in training programs regarding management of long-COVID [192]. Long-COVID involves fatigue and fatigue with exertion as perhaps the most prominent reported symptoms (leading, indeed, to comparisons of long-COVID to chronic fatigue syndrome/myalgic encephalomyelitis—and, indeed, some patients meet criteria for this condition) [23,193,194,195]. However, statins were reported—in the only RCT we are aware of that assessed this—to significantly increase fatigue and fatigue with exertion (compared to placebo) [18]. Muscle weakness is also a significant feature of long-COVID [22] that can also be engendered by statins [19,21] through mitochondrial [31,196], oxidative [102], and autoimmune mechanisms [10,11,12,138,139,140,141,142]—mechanisms shared with COVID-19 [114,197,198,199,200,201,202,203]. Long-COVID also commonly involves reported cognitive compromise [194,195]: this is also a reported adverse effect of statins [20,204] and, indeed, is the subject of an FDA label change [205]. Shortness of breath (dyspnea) arises with long-COVID [23] and can be a feature in statin myopathy: both respiratory muscle myopathy [206] and, potentially, simply impaired mitochondrial respiration [207] may be contributors. The two randomized trials of statin use that assessed cognitive function, focusing on non-elderly and pre-trained participants of cognitive testing (each reducing variance and change variance and enhancing statistical power; there are large and highly variable training effects with cognitive tests [208]) showed significant (adverse) cognitive change when on statins relative to the placebo [209,210]. Weakness is also reported as a long-COVID symptom in some and has been found to be worsened when on statins (versus placebo) in key at-risk groups—including those with any baseline weakness (as may be present with long-COVID) who exercise regularly (as may be advocated with long-COVID) and in older women [19]. These findings were replicated across each of two assessed statins compared to the placebo, so are unlikely to represent chance findings.

Statins lead to dose-dependent reductions in CoQ10 [163,211,212], which plays a crucial role in cellular energy production. Both mitochondrial impairment and oxidative stress are considerations in COVID-19 [114]; chronic fatigue has been tied to mitochondrial impairment [213]; oxidative stress and mitochondrial impairment each can arise with statins and are tied to statin adverse effects [31,101,102]). Of note, long-COVID may be disproportionately common in the setting of diabetes: statins can increase risk of diabetes [214,215] and diabetes also increases risk of statin complications [31].

In the post-COVID-19 setting, benefits of lowered testosterone to viral cell entry (Section 8) may lose relevance, while statin-induced testosterone depressions continue to have adverse implications for fatigue [216,217,218], muscle strength [219,220,221], and cognition [222], among other central features of long-COVID. An exception arises for instances in which there is persistence of COVID-19 virus, which is indeed one of four identified patterns of post-COVID syndrome, also known as long-COVID [159,160].

We suggest that statin use (or at least statin initiation targeted to COVID-19 mitigation) should be avoided in long-COVID unless and until RCT data affirming the benefit become available.

## 13. Summary

In a comprehensive meta-analysis of randomized trials, there is a conspicuous absence of evidence suggesting that statins reduce infection or infection-related mortality [52,53]. This contradicts the apparently favorable outcomes linked to statin use in observational studies, a discrepancy that can be attributed substantially to the phenomena of “healthy-user” and “healthy-tolerator” effects. Similar spurious associations have also been observed in COVID-19 studies. 

Many observational studies report a positive correlation between statin use and improved COVID-19 outcomes. However, these findings are uncorroborated by randomized trials. Severe infections are often accompanied by significant cholesterol reduction, thereby reducing the likelihood of receiving adjunctive statins (confounding by indication). This dynamic could compound the potential for misleadingly favorable observational associations between statin use and better COVID-19 outcomes.

Statins are known for their anti-inflammatory effects [2], which could theoretically provide some benefit in the context of COVID-19. However, statins may also exert adverse effects on COVID-19 incidence and outcomes: LDL cholesterol, a central transporter of key fat-soluble nutrients, plays a vital role in viral defense, antioxidation, and energy support. Such considerations may offset statin mechanisms that many have presumed would confer infection benefit.

Groups that are particularly vulnerable to severe COVID-19 outcomes—including the elderly and individuals with obesity, hypertension, and diabetes—are also at heightened risk for adverse effects from statin use. Although statins have known anti-inflammatory effects and confer protection against cardiovascular outcomes in many settings, the absence of infection protection and the potential promotion of cardiac as well as skeletal muscle compromise, which may worsen COVID-19 outcomes [109], create significant concerns in relation to statin use in COVID-19 (where CHF is both a risk factor and an adverse outcome [223]).

Against these considerations, use of statins for COVID-19 infection has been endorsed in position pieces that have achieved broad circulation. (See Appendix A.) Indeed, a widely circulated MGH document [1] that endorsed statins for treatment of COVID-19 cited observational studies that had shown an apparent association between statin use and better infection outcomes but had not referenced the meta-analysis of randomized trials that showed no suggestion of a causal relationship of statins to better infection outcomes [52,53]. Recommendations should avoid appearing to promote off-label use of a prescription drug class that has prospects for material harm, in the absence of clear overriding benefit in RCTs.

Perhaps most importantly, hospitalized COVID-19 patients have shown high reported rates of rhabdomyolysis, a well-known complication of statin use. Moreover, such patients have had high reported mortality. Given this, the addition of an agent—like statins—that magnifies risk of rhabdomyolysis may be imprudent, particularly in absence of unequivocal RCT evidence countermanding clear prospects for harm.

Regarding statin use in long-COVID, statins have potential to cause core long-COVID symptoms, including fatigue (increased on average, in RCT evidence) [18], as well as muscle [21,31,196] and cognitive [20,204] problems. Mechanisms of toxicity engaged by both statins and the COVID spike protein include oxidative stress and mitochondrial impairment, which may underlie many or most long-COVID symptoms, and some individuals sensitive to these effects from one of the exposures may be sensitive to the effects from both. At present, there is no evidence that the benefits of statins in the long-COVID setting convincingly trump their harms.

## 14. Conclusions

Many of the very patient characteristics that magnify risk with COVID-19 and might prompt physicians to consider unproven statin therapy also magnify risks with statins. Problematically, COVID-19 may itself increase risks of serious and fatal adverse outcomes from statins, and statins might themselves increase risks of serious and fatal adverse outcomes from COVID-19. This extends to rhabdomyolysis—for which reported risks have been high in hospitalized COVID-19 patients.

Given the evident risks and the absence of clear benefit of statins to COVID-19, we suggest clinicians use caution in initiating statins for COVID-19 or long-COVID, unless/until RCT data clearly support this intervention.

## Figures and Tables

**Figure 1 jcm-12-04659-f001:**
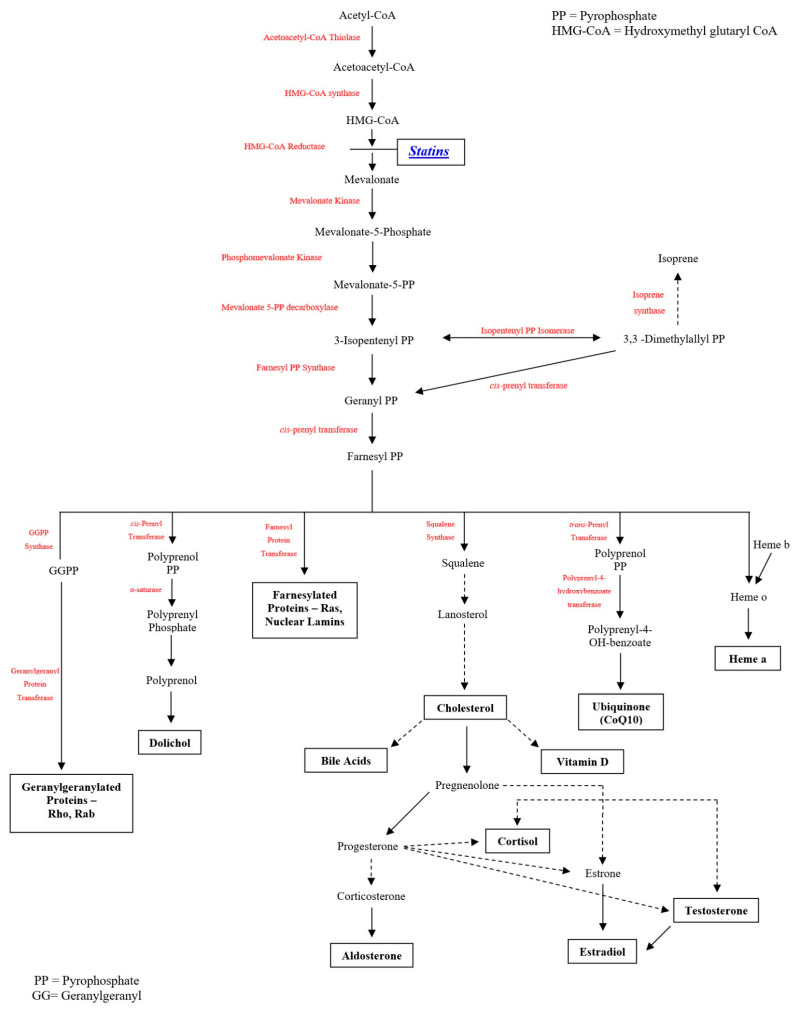
Mevalonate Pathway Inhibition by Statins (HMG-CoA Reductase Inhibitors).

**Table 2 jcm-12-04659-t002:** Randomized controlled trials and meta-analysis of statin use in the COVID-19 setting.

First Author (Year)	N	Population	Statin (Dose)	Outcomes	Measure	Results	95% CI	*p*
Ghafoori et al. (2022) [72]	156	Adult hospitalized patients with COVID-19, in Bojnourd city	Atorvastatin (20 mg)	Mean hospitalization days	Number of days	7.72 days (statin) vs. 5.06 days (placebo)	N/A	*p* = 0.001
				Frequency of hospitalization in the ICU	Percent frequency	18.4% (statin) vs. 1.3% (placebo)		*p* = 0.001
INSPIRATION-S Investigators (2022) [73]	587	Adult hospitalized patients with COVID-19, admitted to the ICU, in Iran	Atorvastatin (20 mg)	Composite of venous or arterial thrombosis, treatment with extracorporeal membrane oxygenation, or all-cause mortality	OR	0.84	0.58–1.21	N/A
Davoodi et al. (2021) [74]	40	Adult hospitalized patients with COVID-19, in Iran	Atorvastatin (40 mg)	Primary outcome: Duration of hospitalization	Days	9.75 ± 2.29 (control) vs. 7.95 ± 2.04 (statin)	N/A	*p* = 0.012
Ghati et al. (2022) [75]	900	Adult hospitalized patients with COVID-19, in Jhajjar, Haryana (India)	Atorvastatin (40 mg)	Primary outcome: “clinical deterioration to WHO Ordinal Scale for Clinical Improvement ≥ 6”.	Rate of outcome	3.2%	N/A	N/A
					HR	1.0	0.41–2.46	*p* = 0.99
Meta-analysis of above studies								
Xavier et al. (2023) [77]	1231	Varies by study. See above	Varies by study. See above	All-cause mortality	OR	0.96	0.61–1.51	*p* = 0.86
				Duration of hospitalization	Mean difference	0.21	−1.74–2.16	*p* = 0.83
				ICU admission	OR	3.31	0.13–87.1	*p* = 0.47
				Need for mechanical ventilation	OR	1.03	0.36–2.94	*p* = 0.95

OR = odds ratio; HR = hazard ratio.

## Data Availability

No new data were created or analyzed in this study. Data sharing is not applicable to this article.

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
