# Peer review of "Statin Use in Relation to COVID-19 and Other Respiratory Infections: Muscle and Other Considerations"

_jcm, 2023, doi:10.3390/jcm12144659_

Round 1
Reviewer 1 Report
This is an extremely well-written, comprehensive assessment of whether statin use should be advocated or patients diagnosed with Covid-19. The authors addressed the extensive and opposing literature on statin use outcomes in observational versus interventional research on Covid-19 and other infectious diseases. The authors' assessments are highly objective and thorough in their in-depth analysis of potential statin use in Covid-19 treatment. They make a strong case against the use of statins as a Covid-19 treatment based on multiple levels of analysis, including an extensive review of adverse effects of statins, which may even be exacerbated in the context of Covid-19 illness.
Author Response
Comment: This is an extremely well-written, comprehensive assessment of whether statin use should be advocated or patients diagnosed with Covid-19. The authors addressed the extensive and opposing literature on statin use outcomes in observational versus interventional research on Covid-19 and other infectious diseases. The authors' assessments are highly objective and thorough in their in-depth analysis of potential statin use in Covid-19 treatment. They make a strong case against the use of statins as a Covid-19 treatment based on multiple levels of analysis, including an extensive review of adverse effects of statins, which may even be exacerbated in the context of Covid-19 illness.
Response: We thank Reviewer 1 for their thoughtful and generous comments.
Reviewer 2 Report
2. Statins
The pleitropic effects of statins do not only affect the HMG-CoA reductase inhibition pathway (Figure 1). In Chapter 2 these should be mentioned (possibly with potential statin targets). It must be explained why the mevalonate pathway was exclusively processed in the case of the Covid 19 infection. The resolution of Figure 1 was not suitable for me.
4. Statins and COVID-19 Outcomes
It would be easier to interpret and compare if the results were summarized in a table (in the columns, for example, number of participants, control patient treatment, type of statin, daily dose, main finding).
5. Cholesterol and infection
This chapter is unfinished, not thought through by the authors. Many studies have examined the relationship between the outcome of the Covid 19 infection and cholesterol levels. LDLc is mentioned in the manuscript, but the severity of Covid 19 infection and HDLc/cholesterol was also investigated. I recommend that the chapter be reworked by processing meta-analysis data, similar to chapter 2. Chapter 12 should also discuss the cholesterol-lowering effect of statins and the protective effect of cholesterol against Covid-19.
Author Response
Section 2. Statins
Comment: The pleitropic effects of statins do not only affect the HMG-CoA reductase inhibition pathway (Figure 1). In Chapter 2 these should be mentioned (possibly with potential statin targets). It must be explained why the mevalonate pathway was exclusively processed in the case of the Covid 19 infection. The resolution of Figure 1 was not suitable for me.
Response: Thank you for your comment about pleiotropic effects. Obviously, this is a potentially big topic, but our focus is on mevalonate pathway effects, and mitochondrial and oxidative stress, in part because these likely substantially underlie such pleiotropic effects. Oxidative stress drives endothelial dysfunction, so that when statins are antioxidant, endothelial function improvements are anticipated. Oxidative stress promotes apoptosis, which drives inflammation and coagulation activation, to other so-called pleiotropic effects of statins. In fact, recent data we procured assessing mitochondrial function on biopsy and inflammation in a specific group (documented high risk of heart disease, and with known elevations in inflammation and coagulation activation), we found that it was that the increased inflammation was driven by impaired mitochondrial fatty acid oxidation (which is known specifically to drive apoptosis). So although a textbook can easily be written on pleiotropic effects, the mechanisms that underlie them and the consequences of them, we think the relative focus on mitochondrial and oxidative mechanisms (including antioxidant transport by cholesterol) is justified. We agree that it is desirable to point out that statins have many other reported effects, that these are influenced by mechanisms including but not necessarily limited to mitochondrial and oxidative ones, but a full discussion of possible relations of all such mechanisms of statins to COVID-19 outcomes is beyond the scope of this manuscript. That said, the following text has been added:
“Statins have been reported to promote multiple pleiotropic mechanisms, including benefits to endothelial function, inflammation, coagulation activation, as well as antioxidation. However, antioxidation itself may be responsible for much or most of these benefits, since antioxidation supports endothelial function, and reduces apoptosis that in turn promotes inflammation and coagulation activation. (In fact, inflammation and coagulation activation were strong predictors of severe COVID-19 outcomes, and this may in part arise from the fact that these reflect oxidatively-mediated cell death.) A full discussion of pleiotropic effects of statins, the factors that underlie these, and the possible contribution of these to effects of statins on COVID-19, are outside the scope of this paper: The absence of randomized trial support for net statin benefit in treating patients with COVID-19 materially diminishes the imperative for detailed exposition of these pleiotropic mechanisms. Our focus is on key documented mechanisms with direct relevance for statins, for COVID-19, and for the interaction between these.”
A brief legend has also been added to Figure 1, describing some pathway elements.
Section 4. Statins and COVID-19 Outcomes
Comment: It would be easier to interpret and compare if the results were summarized in a table (in the columns, for example, number of participants, control patient treatment, type of statin, daily dose, main finding).
Response: Done.
Section 5. Cholesterol and infection
Comment: This chapter is unfinished, not thought through by the authors. Many studies have examined the relationship between the outcome of the Covid 19 infection and cholesterol levels. LDLc is mentioned in the manuscript, but the severity of Covid 19 infection and HDLc/cholesterol was also investigated. I recommend that the chapter be reworked by processing meta-analysis data, similar to chapter 2. Chapter 12 should also discuss the cholesterol-lowering effect of statins and the protective effect of cholesterol against Covid-19.
Response: The reviewer makes an excellent point, that cholesterol is not defined only by LDL, but also HDL. We agree that mention of HDL is warranted. We have added the following text: “Cholesterol declines more strongly with greater COVID-19 severity, as shown in multiple studies and meta-analyses, including for total cholesterol, LDL, and HDL cholesterol. These drops in lipids not only reflect severity of disease, but because both LDL and HDL are involved in transporting critical fat-soluble antioxidants, and HDL also transports substances like paraoxonase with antioxidant (and anti-SARS-CoV-2 properties) properties, these reductions might not just reflect, but may conceivably contribute to worse COVID-19-associated outcomes. Statin use clinically is typically tied to/driven by levels of LDL cholesterol, which reproducibly respond to statin therapy. HDL cholesterol is often, on average, neutrally influenced by statin therapy. For these reasons, although HDL is of vital importance as a reflector of and contributor to oxidative stress defense potential, it is not the recipient of equal attention in the discussion here.”
Reviewer 3 Report
This systematic review and meta-analysis were to analyze the Statin Use in relation to COVID-19 and Other Respiratory Infections. The content of the review is significant, the presentation is pretty good and relevant to the field. It is very interesting: higher cholesterol was tied to lower risk of death from respiratory diseases.
For the integrity of the review, is there any new studies after new drugs, like Paxlovid, used in COVID-19 treatment?
There are other minor revisions may need:
1. Figure 1 is so fuzzy and blurred. Replace it with a high-definition one please.
2. Line 400: Change vis-à-vis into English.
3. Line 449: focused in on. Delete the “in”
4. Line 466: Replace long-Covid with long-COVID
5. Line 481: Change large into regular font.
6. Line 493: Change may into regular font.
7. Line 537: Change casual into regular font.
English language is fine.
Author Response
Comment: For the integrity of the review, is there any new studies after new drugs, like Paxlovid, used in COVID-19 treatment?
Response: We thank the reviewer for their thoughtful comments. If the reviewer is wondering whether there are statin RCTs in Paxlovid-treated patients, we have not identified any. However, the reviewer makes an excellent point – there is evidence of drug interactions between Paxlovid and statins that might worsen the risk-benefit balance of statins in the COVID-19 setting. We have added the following sentence to Section 4:
“Treatments for COVID-19 may also modify statins’ risk-benefit balance, through drug interactions. For instance, nirmatrelvir/ritonavir (“Paxlovid”), by inhibiting the CYP3A4 metabolizing pathway, may heighten risks associated with statins in proportion to the degree to which they are metabolized through that pathway: this especially affects simvastatin and lovastatin, which should not be coadministered, with potential interactions for atorvastatin and rosuvastatin for which close monitoring and potential dose adjustment are advised.”
Comment: There are other minor revisions may need:
- Figure 1 is so fuzzy and blurred. Replace it with a high-definition one please.
Response: We thank the reviewer for pointing this out. The image resolution seemed to be clear on our end upon our initial submission, but we agree the image shown to us on the portal appears fuzzy. We have endeavored to fix this.
- Line 400: Change vis-à-vis into English.
Response: We have changed the wording to “in relation to.” We note, however, that, to our knowledge, every mainstream English language dictionary contains the term vis-à-vis. It is one of innumerable loanwords that is commonly used in the English language.
- Line 449: focused in on. Delete the “in”
Response: Done.
- Line 466: Replace long-Covid with long-COVID
Response: Done.
- Line 481: Change large into regular font.
Response: Done.
- Line 493: Change may into regular font.
Response: Done.
- Line 537: Change casual into regular font.
Response: Done.
We thank the reviewer for their thoughtful and constructive comments.
Reviewer 4 Report
Here are my suggestions :
1. Reference missing on line no. 46 line no. 2 "Rhabdomyolysis and autoimmune necrotizing myositis specifically have occurred with COVID-19 as they can occur with statins. "
2. Can you please redraft the sentence as it's not clear what the author aims to 8 convey on line page 338?
3. Epidemiological surveillance of the COVID-19 pandemic has demonstrated a male vulnerability to morbidity and mortality, despite similar infection rates between the two sexes (DOI: 10.1016/j.micinf.2021.104850). You should mention this in your discussion or manuscript. It correlates with your arguments.
4. The role of covid 19 and liver dysfunction (DOI: 10.1016/j.micinf.2021.104850, ) is well documented, and similarly, the role of Statin in hepatoxicity and statin-induces liver injury. The author should mention it in the manuscript for an overall view.
There are a few typos. The author may look into and rectify them.
Author Response
Comment: Reference missing on line no. 46 line no. 2 “Rhabdomyolysis and autoimmune necrotizing myositis specifically have occurred with COVID-19 as they can occur with statins.”
Response: These references had been included later, but have been added here.
Comment: Can you please redraft the sentence as it’s not clear what the author aims to 8 convey on line page 338?
Response: In the version of the manuscript downloaded from the portal, line 338 appears to be “They are particularly apt to do so in older age and presence of metabolic syndrome features” preceded by the sentence “Statins can increase muscle…” Our best guess is that the word “They” (referring to statins) and “apt to do so” were viewed as unclear. We have changed the sentence to read “Statins are particularly apt to produce such muscle compromise in older age and presence of metabolic syndrome features.” We hope this clarifies the intent of the sentence.
Comment: Epidemiological surveillance of the COVID-19 pandemic has demonstrated a male vulnerability to morbidity and mortality, despite similar infection rates between the two sexes (DOI: 10.1016/j.micinf.2021.104850). You should mention this in your discussion or manuscript. It correlates with your arguments.
Response: We thank the reviewer for pointing out that we have omitted to mention the sex difference although we had cited an article titled “Sex differences in COVID-19: the role of androgens in disease severity and progression.” We have made this point more explicit, and added the proposed citation, with the sentence now reading: “Since COVID-19 entry into cells is androgen dependent (believed to contribute to the male preponderance of severe COVID-19), this mechanism might confer benefit in relation to acute COVID-19 [Mohamed et al., Pegiou et al.].”
Comment: The role of covid 19 and liver dysfunction (DOI: 10.1016/j.micinf.2021.104850) is well documented, and similarly, the role of Statin in hepatoxicity and statin-induces liver injury. The author should mention it in the manuscript for an overall view.
Response: This is a very relevant point – we are focusing on muscle disproportionately to hepatotoxicity because muscle complications of statins have been reported to be considerably more clinically prevalent, but this reviewer is correct that a more complete discussion should include hepatotoxicity considerations. We have added a new section 8 called “Statins and COVID-19: Effects on Liver,” with relevant citations.
We thank the reviewer for their thoughtful and constructive comments.
Reviewer 5 Report
This is an interesting review of the topic. I have just some comments/suggestions to the authors.
Despite a large number of references cited (not all of them necessary) the authors should comment and cite the first published article associating the effects of statins on Sars-Cov-2 virus: Reiner Ž, et al. Statins and the COVID-19 main protease: in silico evidence on direct interaction.Arch Med Sci. 2020 Apr 25;16(3):490-496.
The authors might also comment and cite an important paper: Kouhpeikar H, et al.Statin Use in COVID-19 Hospitalized Patients and Outcomes: A Retrospective Study.
Some typos and small grammar mistakes shouldd be improved.
Author Response
Comment: Despite a large number of references cited (not all of them necessary) the authors should comment and cite the first published article associating the effects of statins on Sars-Cov-2 virus: Reiner Ž, et al. Statins and the COVID-19 main protease: in silico evidence on direct interaction.Arch Med Sci. 2020 Apr 25;16(3):490-496.
Response: We thank the reviewer for their kind suggestion. Although we have elected to focus almost exclusively on human data, with very limited inclusion of a couple of animal references, we will add reference to this in silico finding. This has been positioned in the first paragraph, adding the reference to the sentence “Points in favor of statins include their anti-inflammatory effects and reduction of cardiovascular outcomes, outcomes that can arise in association with COVID-19 [Reiner et al.]” and inserting a new parenthesized sentence stating, “Among additional hypothesized mechanisms of benefit include in silico evidence that some statins may affect the virus itself [Reiner et al.].”
Comment: The authors might also comment and cite an important paper: Kouhpeikar H, et al.Statin Use in COVID-19 Hospitalized Patients and Outcomes: A Retrospective Study.Front Cardiovasc Med. 2022 Feb 24;9:820260.
Response: A sentence has been added in section 4: “(Subsequent observational studies of statin use in hospitalized COVID-19 patients show analogous favorable associations to severe and fatal COVID-19 outcomes [Kouhpeikar et al.].).”
We thank the reviewer for their thoughtful and constructive comments.
Round 2
Reviewer 5 Report
Since the authors have made all the suggested changes I have no further comments/suggestions.